# On the Statistical Consistency of Plug-in Classifiers for Non-decomposable Performance Measures

**Harikrishna Narasimhan[†], Rohit Vaish[†], Shivani Agarwal**
Department of Computer Science and Automation
Indian Institute of Science, Bangalore – 560012, India
{harikrishna, rohit.vaish, shivani}@csa.iisc.ernet.in

## Abstract

We study consistency properties of algorithms for non-decomposable performance measures that cannot be expressed as a sum of losses on individual data points, such as the F-measure used in text retrieval and several other performance measures used in class imbalanced settings. While there has been much work on designing algorithms for such performance measures, there is limited understanding of the theoretical properties of these algorithms. Recently, Ye et al. (2012) showed consistency results for two algorithms that optimize the F-measure, but their results apply only to an idealized setting, where precise knowledge of the underlying probability distribution (in the form of the 'true' posterior class probability) is available to a learning algorithm. In this work, we consider plug-in algorithms that learn a classifier by applying an empirically determined threshold to a suitable 'estimate' of the class probability, and provide a general methodology to show consistency of these methods for any non-decomposable measure that can be expressed as a continuous function of true positive rate (TPR) and true negative rate (TNR), and for which the Bayes optimal classifier is the class probability function thresholded suitably. We use this template to derive consistency results for plug-in algorithms for the F-measure and for the geometric mean of TPR and precision; to our knowledge, these are the first such results for these measures. In addition, for continuous distributions, we show consistency of plug-in algorithms for any performance measure that is a continuous and monotonically increasing function of TPR and TNR. Experimental results confirm our theoretical findings.

## 1 Introduction

In many real-world applications, the performance measure used to evaluate a learning model is non-decomposable and cannot be expressed as a summation or expectation of losses on individual data points; this includes, for example, the F-measure used in information retrieval [1], and several combinations of the true positive rate (TPR) and true negative rate (TNR) used in class imbalanced classification settings [2–5] (see Table 1). While there has been much work in the last two decades in designing learning algorithms for such performance measures [6–14], our understanding of the statistical consistency of these methods is rather limited. Recently, Ye et al. (2012) showed consistency results for two algorithms for the F-measure [15] that use the 'true' posterior class probability to make predictions on instances. These results implicitly assume that the given learning algorithm has precise knowledge of the underlying probability distribution (in the form of the true posterior class probability); this assumption does not however hold in most real-world settings.

In this paper, we consider a family of methods that construct a plug-in classifier by applying an empirically determined threshold to a suitable 'estimate' of the class probability (obtained using a model learned from a sample drawn from the underlying distribution). We provide a general method-

---

[†]Both authors contributed equally to this paper.

Table 1: Performance measures considered in our study. Here $\beta \in (0, \infty)$ and $p = \mathbf{P}(y = 1)$. Each performance measure here can be expressed as $\mathcal{P}_D^{\Psi}[h] = \Psi(\text{TPR}_D[h], \text{TNR}_D[h], p)$. The last column contains the assumption on the distribution $D$ under which the plug-in algorithm considered in this work is statistically consistent w.r.t. a performance measure (details in Sections 3 and 5).

| Measure | Definition | Ref. | $\Psi(u, v, p)$ | Assumption on $D$ |
|---|---|---|---|---|
| AM (1-BER) | $(\text{TPR} + \text{TNR})/2$ | [17–19] | $\frac{u+v}{2}$ | Assumption A |
| F$_\beta$-measure | $(1 + \beta^2)/\left(\frac{\beta^2}{\text{Prec}} + \frac{1}{\text{TPR}}\right)$ | [1, 19] | $\frac{(1+\beta^2)pu}{p+\beta^2(pu+(1-p)(1-v))}$ | Assumption A |
| G-TP/PR | $\sqrt{\text{TPR} \cdot \text{Prec}}$ | [3] | $\sqrt{\frac{pu^2}{pu+(1-p)(1-v)}}$ | Assumption A |
| G-Mean (GM) | $\sqrt{\text{TPR} \cdot \text{TNR}}$ | [2, 3] | $\sqrt{uv}$ | Assumption B |
| H-Mean (HM) | $2/\left(\frac{1}{\text{TPR}} + \frac{1}{\text{TNR}}\right)$ | [4] | $\frac{2uv}{u+v}$ | Assumption B |
| Q-Mean (QM) | $1 - ((1 - \text{TPR})^2 + (1 - \text{TNR})^2)/2$ | [5] | $1 - \frac{(1-u)^2+(1-v)^2}{2}$ | Assumption B |

ology to show statistical consistency of these methods (under a mild assumption on the underlying distribution) for any performance measure that can be expressed as a continuous function of the TPR and TNR and the class proportion, and for which the Bayes optimal classifier is the class probability function thresholded at a suitable point. We use our proof template to derive consistency results for the F-measure (using a recent result by [15] on the Bayes optimal classifier for F-measure), and the geometric mean of TPR and precision; to our knowledge, these are the first such results for these performance measures. Using our template, we also obtain a recent consistency result by Menon et al. [16] for the arithmetic mean of TPR and TNR. In addition, we show that for continuous distributions, the optimal classifier for any performance measure that is a continuous and monotonically increasing function of TPR and TNR is necessarily of the requisite thresholded form, thus establishing consistency of the plug-in algorithms for all such performance measures. Experiments on real and synthetic data confirm our theoretical findings, and show that the plug-in methods considered here are competitive with the state-of-the-art SVM$^{\text{perf}}$ method [12] for non-decomposable measures.

**Related Work.** Much of the work on non-decomposable performance measures in binary classification settings has focused on the F-measure; these include the empirical plug-in algorithm considered here [6], cost-weighted versions of SVM [9], methods that optimize convex and non-convex approximations to F-measure [10–14], and decision-theoretic methods that learn a class probability estimate and compute predictions that maximize the expected F-measure on a test set [7–9]. While there has been considerable amount of work on consistency of algorithms for univariate performance measures [16, 20–22], theoretical results on non-decomposable measures have been limited to characterizing the Bayes optimal classifier for F-measure [15, 23, 24], and some consistency results for F-measure for certain idealized versions of the empirical plug-in and decision theoretic methods that have access to the true class probability [15]. There has also been some work on algorithms that optimize F-measure in multi-label classification settings [25, 26] and consistency results for these methods [26, 27], but these results do not apply to the binary classification setting that we consider here; in particular, in a binary classification setting, the F-measure that one seeks to optimize is a single number computed over the entire training set, while in a multi-label setting, the goal is to optimize the mean F-measure computed over multiple labels on individual instances.

**Organization.** We start with some preliminaries in Section 2. Section 3 presents our main result on consistency of plug-in algorithms for non-decomposable performance measures that are functions of TPR and TNR. Section 4 contains application of our proof template to the AM, F$_\beta$ and G-TP/PR measures, and Section 5 contains results under continuous distributions for performance measures that are monotonic in TPR and TNR. Section 6 describes our experimental results on real and synthetic data sets. Proofs not provided in the main text can be found in the Appendix.

## 2 Preliminaries

**Problem Setup.** Let $\mathcal{X}$ be any instance space. Given a training sample $S = ((x_1, y_1), \ldots, (x_n, y_n)) \in (\mathcal{X} \times \{\pm 1\})^n$, our goal is to learn a binary classifier $\widehat{h}_S : \mathcal{X} \to \{\pm 1\}$ to make predictions for new instances drawn from $\mathcal{X}$. Assume all examples (both training and test) are drawn iid according to some unknown probability distribution $D$ on $\mathcal{X} \times \{\pm 1\}$. Let $\eta(x) = \mathbf{P}(y = 1|x)$ and $p = \mathbf{P}(y = 1)$ (both under $D$). We will be interested in settings where the performance of $\widehat{h}_S$ is measured via a *non-decomposable performance measure* $\mathcal{P} : \{\pm 1\}^{\mathcal{X}} \to \mathbb{R}_+$, which cannot be expressed as a sum or expectation of losses on individual examples.

**Non-decomposable performance measures.** Let us first define the following quantities associated with a binary classifier $h : \mathcal{X} \to \{\pm 1\}$:

| | | | |
|---|---|---|---|
| True Positive Rate / Recall | $\text{TPR}_D[h]$ | $=$ | $\mathbf{P}\big(h(x) = 1 \,\vert\, y = 1\big)$ |
| True Negative Rate | $\text{TNR}_D[h]$ | $=$ | $\mathbf{P}\big(h(x) = -1 \,\vert\, y = -1\big)$ |
| Precision | $\text{Prec}_D[h]$ | $=$ | $\mathbf{P}\big(y = 1 \,\vert\, h(x) = 1\big) = \frac{p\,\text{TPR}_D[h]}{p\,\text{TPR}_D[h] + (1-p)(1 - \text{TNR}_D[h])}.$ |

In this paper, we will consider non-decomposable performance measures that can be expressed as a function of the TPR and TNR and the class proportion $p$. Specifically, let $\Psi : [0,1]^3 \to \mathbb{R}_+$; then the $\Psi$-performance of $h$ w.r.t. $D$, which we will denote as $\mathcal{P}_D^\Psi[h]$, will be defined as:

$$\mathcal{P}_D^\Psi[h] = \Psi(\text{TPR}_D[h],\, \text{TNR}_D[h],\, p).$$

For example, for $\beta > 0$, the $F_\beta$-measure of $h$ can be defined through the function $\Psi_{F_\beta} : [0,1]^3 \to \mathbb{R}_+$ given by $\Psi_{F_\beta}(u, v, p) = \frac{(1+\beta^2)pu}{p + \beta^2(pu + (1-p)(1-v))}$, which gives $\mathcal{P}_D^{F_\beta}[h] = (1 + \beta^2)/\left(\frac{\beta^2}{\text{Prec}_D[h]} + \frac{1}{\text{TPR}_D[h]}\right)$. Table 1 gives several examples of non-decomposable performance measures that are used in practice. We will also find it useful to consider empirical versions of these performance measures calculated from a sample $S$, which we will denote as $\widehat{\mathcal{P}}_S^\Psi[h]$:

$$\widehat{\mathcal{P}}_S^\Psi[h] = \Psi(\widehat{\text{TPR}}_S[h],\, \widehat{\text{TNR}}_S[h],\, \widehat{p}_S), \tag{1}$$

where $\widehat{p}_S = \frac{1}{n} \sum_{i=1}^{n} \mathbf{1}(y_i = 1)$ is an empirical estimate of $p$, and

$$\widehat{\text{TPR}}_S[h] = \frac{1}{\widehat{p}_S n} \sum_{i=1}^{n} \mathbf{1}(h(x_i) = 1,\, y_i = 1); \quad \widehat{\text{TNR}}_S[h] = \frac{1}{(1 - \widehat{p}_S)n} \sum_{i=1}^{n} \mathbf{1}(h(x_i) = -1,\, y_i = -1)$$

are the empirical TPR and TNR respectively.[1]

**$\Psi$-consistency.** We will be interested in the optimum value of $\mathcal{P}_D^\Psi$ over all classifiers:

$$\mathcal{P}_D^{\Psi,*} = \sup_{h : \mathcal{X} \to \{\pm 1\}} \mathcal{P}_D^\Psi[h].$$

In particular, one can define the $\Psi$-regret of a classifier $h$ as:

$$\text{regret}_D^\Psi[h] = \mathcal{P}_D^{\Psi,*} - \mathcal{P}_D^\Psi[h].$$

A learning algorithm is then said to be $\Psi$-consistent if the $\Psi$-regret of the classifier $\widehat{h}_S$ output by the algorithm on seeing training sample $S$ converges in probability to 0:[2]

$$\text{regret}_D^\Psi[\widehat{h}_S] \xrightarrow{P} 0.$$

**Class of Threshold Classifiers.** We will find it useful to define for any function $f : \mathcal{X} \to [0,1]$, the set of classifiers obtained by assigning a threshold to $f$: $\mathcal{T}_f = \{\text{sign} \circ (f - t) \,\vert\, t \in [0,1]\}$, where $\text{sign}(u) = 1$ if $u > 0$ and $-1$ otherwise. For a given $f$, we shall also define the thresholds corresponding to maximum population and empirical measures respectively (when they exist) as:

$$t_{D,f,\Psi}^* \in \underset{t \in [0,1]}{\text{argmax}} \; \mathcal{P}_D^\Psi[\text{sign} \circ (f - t)]; \quad \widehat{t}_{S,f,\Psi} \in \underset{t \in [0,1]}{\text{argmax}} \; \widehat{\mathcal{P}}_S^\Psi[\text{sign} \circ (f - t)].$$

**Plug-in Algorithms and Result of Ye et al. (2012).** In this work, we consider a family of plug-in algorithms, which divide the input sample $S$ into samples $(S_1, S_2)$, use a suitable class probability estimation (CPE) algorithm to learn a class probability estimator $\widehat{\eta}_{S_1} : \mathcal{X} \to [0,1]$ from $S_1$, and output a classifier $\widehat{h}_S(x) = \text{sign}(\widehat{\eta}_{S_1}(x) - \widehat{t}_{S_2, \widehat{\eta}_{S_1}, \Psi})$, where $\widehat{t}_{S_2, \widehat{\eta}_{S_1}, \Psi}$ is a threshold that maximizes the empirical performance measure on $S_2$ (see Algorithm 1). We note that this approach is different from the idealized plug-in method analyzed by Ye et al. (2012) in the context of F-measure optimization, where a classifier is learned by assigning an empirical threshold to the 'true' class probability function $\eta$ [15]; the consistency result therein is useful only if precise knowledge of $\eta$ is available to a learning algorithm, which is not the case in most practical settings.

**$L_1$-consistency of a CPE algorithm.** Let $\mathcal{C}$ be a CPE algorithm, and for any sample $S$, denote $\widehat{\eta}_S = \mathcal{C}(S)$. We will say $\mathcal{C}$ is $L_1$-consistent w.r.t. a distribution $D$ if $\mathbf{E}_x\big[\big\vert \widehat{\eta}_S(x) - \eta(x) \big\vert\big] \xrightarrow{P} 0$.

**Algorithm 1** Plug-in with Empirical Threshold for Performance Measure $\mathcal{P}^{\Psi} : 2^{\mathcal{X}} \to \mathbb{R}_+$

---

1: **Input:** $S = ((x_1, y_1), \ldots, (x_n, y_n)) \in (\mathcal{X} \times \{\pm 1\})^n$
2: **Parameter:** $\alpha \in (0, 1)$
3: Let $S_1 = ((x_1, y_1), \ldots, (x_{n_1}, y_{n_1})), S_2 = ((x_{n_1+1}, y_{n_1+1}), \ldots, (x_n, y_n))$, where $n_1 = \lceil n\alpha \rceil$
4: Learn $\widehat{\eta}_{S_1} = \mathcal{C}(S_1)$, where $\mathcal{C} : \cup_{n=1}^{\infty} (\mathcal{X} \times \{\pm 1\})^n \to [0, 1]^{\mathcal{X}}$ is a suitable CPE algorithm
5: $\widehat{t}_{S_2, \widehat{\eta}_{S_1}, \Psi} \in \underset{t \in [0,1]}{\operatorname{argmax}} \ \widehat{\mathcal{P}}_{S_2}^{\Psi}[\operatorname{sign} \circ (\widehat{\eta}_{S_1} - t)]$
6: **Output:** Classifier $\widehat{h}_S(x) = \operatorname{sign}(\widehat{\eta}_{S_1}(x) - \widehat{t}_{S_2, \widehat{\eta}_{S_1}, \Psi})$

---

## 3   A Generic Proof Template for $\Psi$-consistency of Plug-in Algorithms

We now give a general result for showing consistency of the plug-in method in Algorithm 1 for any performance measure that can be expressed as a continuous function of TPR and TNR, and for which the Bayes optimal classifier is obtained by suitably thresholding the class probability function.

**Assumption A.** We will say that a probability distribution $D$ on $\mathcal{X} \times \{\pm 1\}$ satisfies Assumption A w.r.t. $\Psi$ if $t^*_{D, \eta, \Psi}$ exists and is in $(0, 1)$, and the cumulative distribution functions of the random variable $\eta(x)$ conditioned on $y = 1$ and on $y = -1$, $\mathbf{P}(\eta(x) \leq z \mid y = 1)$ and $\mathbf{P}(\eta(x) \leq z \mid y = -1)$, are continuous at $z = t^*_{D, \eta, \Psi}$.[3]

Note that this assumption holds for any distribution $D$ for which $\eta(x)$ conditioned on $y = 1$ and on $y = -1$ is continuous, and also for any $D$ for which $\eta(x)$ conditioned on $y = 1$ and on $y = -1$ is mixed, provided the optimum threshold $t^*_{D, \eta, \Psi}$ for $\mathcal{P}^{\Psi}$ exists and is not a point of discontinuity.

Under the above assumption, and assuming that the CPE algorithm used in Algorithm 1 is $L_1$-consistent (which holds for any algorithm that uses a regularized empirical risk minimization of a proper loss [16, 28]), we have our main consistency result.

**Theorem 1** ($\Psi$-consistency of Algorithm 1). Let $\Psi : [0, 1]^3 \to \mathbb{R}_+$ be continuous in each argument. Let $D$ be a probability distribution on $\mathcal{X} \times \{\pm 1\}$ that satisfies Assumption A w.r.t. $\Psi$, and for which the Bayes optimal classifier is of the form $h^{\Psi, *}(x) = \operatorname{sign} \circ (\eta(x) - t^*_{D, \eta, \Psi})$. If the CPE algorithm $\mathcal{C}$ in Algorithm 1 is $L_1$-consistent, then Algorithm 1 is $\Psi$-consistent w.r.t. $D$.

Before we prove the above theorem, we will find it useful to state the following lemmas. In our first lemma, we state that the TPR and TNR of a classifier constructed by thresholding a suitable class probability estimate at a fixed $c \in (0, 1)$ converge respectively to the TPR and TNR of the classifier obtained by thresholding the true class probability function $\eta$ at $c$.

**Lemma 2** (Convergence of TPR and TNR for fixed threshold). Let $D$ be a distribution on $\mathcal{X} \times \{\pm 1\}$. Let $\widehat{\eta}_S : \mathcal{X} \to [0, 1]$ be generated by an $L_1$-consistent CPE algorithm. Let $c \in (0, 1)$ be an apriori fixed constant such that the cumulative distribution functions $\mathbf{P}(\eta(x) \leq z \mid y = 1)$ and $\mathbf{P}(\eta(x) \leq z \mid y = -1)$ are continuous at $z = c$. We then have

$$\operatorname{TPR}_D[\operatorname{sign} \circ (\widehat{\eta}_S - c)] \xrightarrow{P} \operatorname{TPR}_D[\operatorname{sign} \circ (\eta - c)];$$

$$\operatorname{TNR}_D[\operatorname{sign} \circ (\widehat{\eta}_S - c)] \xrightarrow{P} \operatorname{TNR}_D[\operatorname{sign} \circ (\eta - c)].$$

As a corollary to the above lemma, we have a similar result for $\mathcal{P}^{\Psi}$.

**Lemma 3** (Convergence of $\mathcal{P}^{\Psi}$ for fixed threshold). Let $\Psi : [0, 1]^3 \to \mathbb{R}_+$ be continuous in each argument. Under the statement of Lemma 2, we have

$$\mathcal{P}_D^{\Psi}[\operatorname{sign} \circ (\widehat{\eta}_S - c)] \xrightarrow{P} \mathcal{P}_D^{\Psi}[\operatorname{sign} \circ (\eta - c)].$$

We next state a result showing convergence of the empirical performance measure to its population value for a fixed classifier, and a uniform convergence result over a class of thresholded classifiers.

**Lemma 4** (Concentration result for $\mathcal{P}^{\Psi}$). Let $\Psi : [0, 1]^3 \to \mathbb{R}_+$ be continuous in each argument. Then for any fixed $h : \mathcal{X} \to \{\pm 1\}$, and $\epsilon > 0$,

$$\mathbf{P}_{S \sim D^n} \left( \left| \mathcal{P}_D^{\Psi}[h] - \widehat{\mathcal{P}}_S^{\Psi}[h] \right| \geq \epsilon \right) \to 0 \ \ \text{as} \ \ n \to \infty.$$

**Lemma 5 (Uniform convergence of $\mathcal{P}^{\Psi}$ over threshold classifiers).** Let $\Psi : [0,1]^3 \to \mathbb{R}_+$ be continuous in each argument. For any $f : \mathcal{X} \to [0,1]$ and $\epsilon > 0$,

$$\mathbf{P}_{S \sim D^n}\left( \bigcup_{\theta \in \mathcal{T}_f} \left\{ \left| \mathcal{P}_D^{\Psi}[\theta] - \widehat{\mathcal{P}}_S^{\Psi}[\theta] \right| \geq \epsilon \right\} \right) \to 0 \quad \text{as} \quad n \to \infty.$$

We are now ready to prove our main theorem.

*Proof of Theorem 1.* Recall $t_{D,\eta,\Psi}^* \in \underset{t \in [0,1]}{\operatorname{argmax}} \, \mathcal{P}_D^{\Psi}[\operatorname{sign} \circ (\eta - t)]$ exists by Assumption A. In the following, we shall use $t^*$ in the place of $t_{D,\eta,\Psi}^*$ and $\widehat{t}_{S_2,S_1}$ in the place of $\widehat{t}_{S_2,\widehat{\eta}_{S_1},\Psi}$. We have

$$
\begin{aligned}
\operatorname{regret}_D^{\Psi}[h_S] &= \operatorname{regret}_D^{\Psi}[\operatorname{sign} \circ (\widehat{\eta}_{S_1} - \widehat{t}_{S_2,S_1})] \\
&= \mathcal{P}_D^{\Psi,*} - \mathcal{P}_D^{\Psi}[\operatorname{sign} \circ (\widehat{\eta}_{S_1} - \widehat{t}_{S_2,S_1})] \\
&= \mathcal{P}_D^{\Psi}[\operatorname{sign} \circ (\eta - t^*)] - \mathcal{P}_D^{\Psi}[\operatorname{sign} \circ (\widehat{\eta}_{S_1} - \widehat{t}_{S_2,S_1})],
\end{aligned}
$$

which follows from the assumption on the Bayes optimal classifier for $\mathcal{P}^{\Psi}$. Adding and subtracting empirical and population versions of $\mathcal{P}^{\Psi}$ computed on certain classifiers,

$$
\begin{aligned}
\operatorname{regret}_D^{\Psi}[\operatorname{sign} \circ (\widehat{\eta}_{S_1} - \widehat{t}_{S_2,S_1})] = &\underbrace{\mathcal{P}_D^{\Psi}[\operatorname{sign} \circ (\eta - t^*)] - \mathcal{P}_D^{\Psi}[\operatorname{sign} \circ (\widehat{\eta}_{S_1} - t^*)]}_{\text{term}_1} \\
&+ \underbrace{\mathcal{P}_D^{\Psi}[\operatorname{sign} \circ (\widehat{\eta}_{S_1} - t^*)] - \widehat{\mathcal{P}}_{S_2}^{\Psi}[\operatorname{sign} \circ (\widehat{\eta}_{S_1} - \widehat{t}_{S_2,S_1})]}_{\text{term}_2} \\
&+ \underbrace{\widehat{\mathcal{P}}_{S_2}^{\Psi}[\operatorname{sign} \circ (\widehat{\eta}_{S_1} - \widehat{t}_{S_2,S_1})] - \mathcal{P}_D^{\Psi}[\operatorname{sign} \circ (\widehat{\eta}_{S_1} - \widehat{t}_{S_2,S_1})]}_{\text{term}_3}.
\end{aligned}
$$

We now show convergence for each of the above terms. Applying Lemma 3 with $c = t^*$ (by Assumption A, $t^* \in (0,1)$ and satisfies the necessary continuity assumption), we have $\text{term}_1 \xrightarrow{P} 0$. For $\text{term}_2$, from the definition of threshold $\widehat{t}_{S_2,S_1}$ (see Algorithm 1), we have

$$\text{term}_2 \leq \mathcal{P}_D^{\Psi}[\operatorname{sign} \circ (\widehat{\eta}_{S_1} - t^*)] - \widehat{\mathcal{P}}_{S_2}^{\Psi}[\operatorname{sign} \circ (\widehat{\eta}_{S_1} - t^*)]. \tag{2}$$

Then for any $\epsilon > 0$,

$$
\begin{aligned}
\mathbf{P}_{S \sim D^n}(\text{term}_2 \geq \epsilon) &= \mathbf{P}_{S_1 \sim D^{n_1}, S_2 \sim D^{n-n_1}}(\text{term}_2 \geq \epsilon) \\
&= \mathbf{E}_{S_1}\left[ \mathbf{P}_{S_2 | S_1}(\text{term}_2 \geq \epsilon) \right] \\
&\leq \mathbf{E}_{S_1}\left[ \mathbf{P}_{S_2 | S_1}\left( \left| \mathcal{P}_D^{\Psi}[\operatorname{sign} \circ (\widehat{\eta}_{S_1} - t^*)] - \widehat{\mathcal{P}}_{S_2}^{\Psi}[\operatorname{sign} \circ (\widehat{\eta}_{S_1} - t^*)] \right| \geq \epsilon \right) \right] \\
&\to 0
\end{aligned}
$$

as $n \to \infty$, where the third step follows from Eq. (2), and the last step follows by applying, for a fixed $S_1$, the concentration result in Lemma 4 with $h = \operatorname{sign} \circ (\widehat{\eta}_{S_1} - t^*)$ (given continuity of $\Psi$). Finally, for $\text{term}_3$, we have for any $\epsilon > 0$,

$$
\begin{aligned}
\mathbf{P}_S(\text{term}_3 \geq \epsilon) &= \mathbf{E}_{S_1}\left[ \mathbf{P}_{S_2 | S_1}\left( \widehat{\mathcal{P}}_{S_2}^{\Psi}[\operatorname{sign} \circ (\widehat{\eta}_{S_1} - \widehat{t}_{S_2,S_1})] - \mathcal{P}_D^{\Psi}[\operatorname{sign} \circ (\widehat{\eta}_{S_1} - \widehat{t}_{S_2,S_1})] \geq \epsilon \right) \right] \\
&\leq \mathbf{E}_{S_1}\left[ \mathbf{P}_{S_2 | S_1}\left( \bigcup_{\theta \in \mathcal{T}_{\widehat{\eta}_{S_1}}} \left\{ \left| \widehat{\mathcal{P}}_{S_2}^{\Psi}[\theta] - \mathcal{P}_D^{\Psi}[\theta] \right| \geq \epsilon \right\} \right) \right] \\
&\to 0
\end{aligned}
$$

as $n \to \infty$, where the last step follows by applying the uniform convergence result in Lemma 5 over the class of thresholded classifiers $\mathcal{T}_{\widehat{\eta}_{S_1}} = \{\operatorname{sign} \circ (\widehat{\eta}_{S_1} - t) \,|\, t \in [0,1]\}$ (for a fixed $S_1$). $\qquad \square$

## 4 Consistency of Plug-in Algorithms for AM, $F_\beta$, and G-TP/PR

We now use the result in Theorem 1 to establish consistency of the plug-in algorithms for the arithmetic mean of TPR and TNR, the $F_\beta$-measure, and the geometric mean of TPR and precision.

## 4.1 Consistency for AM-measure

The arithmetic mean of TPR and TNR (AM) or one minus the balanced error rate (BER) is a widely-used performance measure in class imbalanced binary classification settings [17–19]:

$$\mathcal{P}_D^{\mathrm{AM}}[h] = \frac{\mathrm{TPR}_D[h] + \mathrm{TNR}_D[h]}{2}.$$

It can be shown that Bayes optimal classifier for the AM-measure is of the form $h^{\mathrm{AM},*}(x) = \mathrm{sign} \circ (\eta(x) - p)$ (see for example [16]), and that the threshold chosen by the plug-in method in Algorithm 1 for the AM-measure is an empirical estimate of $p$. In recent work, Menon et al. show that this plug-in method is consistent w.r.t. the AM-measure [16]; their proof makes use of a decomposition of the AM-measure in terms of a certain cost-sensitive error and a result of [22] on regret bounds for cost-sensitive classification. We now use our result in Theorem 1 to give an alternate route for showing AM-consistency of this plug-in method.[4]

**Theorem 6 (Consistency of Algorithm 1 w.r.t. AM-measure).** Let $\Psi = \Psi_{\mathrm{AM}}$. Let $D$ be a distribution on $\mathcal{X} \times \{\pm 1\}$ that satisfies Assumption A w.r.t. $\Psi_{\mathrm{AM}}$. If the CPE algorithm $\mathcal{C}$ in Algorithm 1 is $L_1$-consistent, then Algorithm 1 is AM-consistent w.r.t. $D$.

*Proof.* We apply Theorem 1 noting that $\Psi_{\mathrm{AM}}(u, v, p) = (u+v)/2$ is continuous in all its arguments, and that the Bayes optimal classifier for $\mathcal{P}^{\mathrm{AM}}$ is of the requisite thresholded form. □

## 4.2 Consistency for $\mathrm{F}_\beta$-measure

The $\mathrm{F}_\beta$-measure or the (weighted) harmonic mean of TPR and precision is a popular performance measure used in information retrieval [1]:

$$\mathcal{P}_D^{\mathrm{F}_\beta}[h] = \frac{(1+\beta^2)\mathrm{TPR}_D[h]\mathrm{Prec}_D[h]}{\beta^2\mathrm{TPR}_D[h] + \mathrm{Prec}_D[h]} = \frac{(1+\beta^2)p\mathrm{TPR}_D[h]}{p + \beta^2\big(p\mathrm{TPR}_D[h] + (1-p)(1 - \mathrm{TNR}_D[h])\big)},$$

where $\beta \in (0, 1)$ controls the trade-off between TPR and precision. In a recent work, Ye et al. [15] show that the optimal classifier for the $\mathrm{F}_\beta$-measure is the class probability $\eta$ thresholded suitably.

**Lemma 7 (Optimality of threshold classifiers for $\mathrm{F}_\beta$-measure; Ye et al. (2012) [15]).** For any distribution $D$ over $\mathcal{X} \times \{\pm 1\}$ that satisfies Assumption A w.r.t. $\Psi$, the Bayes optimal classifier for $\mathcal{P}^{\mathrm{F}_\beta}$ is of the form $h^{\mathrm{F}_\beta,*}(x) = \mathrm{sign} \circ (\eta(x) - t^*_{D,\eta,\mathrm{F}_\beta})$.

As noted earlier, the authors in [15] show that an idealized plug-in method that applies an empirically determined threshold to the 'true' class probability $\eta$ is consistent w.r.t. the $\mathrm{F}_\beta$-measure . This result is however useful only when the 'true' class probability is available to a learning algorithm, which is not the case in most practical settings. On the other hand, the plug-in method considered in our work constructs a classifier by assigning an empirical threshold to a suitable 'estimate' of the class probability. Using Theorem 1, we now show that this method is consistent w.r.t. the $\mathrm{F}_\beta$-measure.

**Theorem 8 (Consistency of Algorithm 1 w.r.t. $\mathrm{F}_\beta$-measure).** Let $\Psi = \Psi_{\mathrm{F}_\beta}$ in Algorithm 1. Let $D$ be a distribution on $\mathcal{X} \times \{\pm 1\}$ that satisfies Assumption A w.r.t. $\Psi_{\mathrm{F}_\beta}$. If the CPE algorithm $\mathcal{C}$ in Algorithm 1 is $L_1$-consistent, then Algorithm 1 is $\mathrm{F}_\beta$-consistent w.r.t. $D$.

*Proof.* We apply Theorem 1 noting that $\Psi_{\mathrm{F}_\beta}(u, v, p) = \frac{(1+\beta^2)pu}{p+\beta^2(pu+(1-p)(1-v))}$ is continuous in each argument, and that (by Lemma 7) the Bayes optimal classifier for $\mathcal{P}^{\mathrm{F}_\beta}$ is of the requisite form. □

## 4.3 Consistency for G-TP/PR

The geometric mean of TPR and precision (G-TP/PR) is another performance measure proposed for class imbalanced classification problems [3]:

$$\mathcal{P}_D^{\mathrm{G\text{-}TP/PR}}[h] = \sqrt{\mathrm{TPR}_D[h]\mathrm{Prec}_D[h]} = \sqrt{\frac{p\mathrm{TPR}_D[h]^2}{p\mathrm{TPR}_D[h] + (1-p)(1 - \mathrm{TNR}_D[h])}}.$$

We first show that the optimal classifier for G-TP/PR is obtained by thresholding the class probability function $\eta$ at a suitable point; our proof uses a technique similar to the one for the $F_\beta$-measure in [15].

**Lemma 9** (**Optimality of threshold classifiers for G-TP/PR**). For any distribution $D$ on $\mathcal{X} \times \{\pm 1\}$ that satisfies Assumption A w.r.t. $\Psi$, the Bayes optimal classifier for $\mathcal{P}^{\text{G-TP/PR}}$ is of the form $h^{\text{G-TP/PR},*}(x) = \text{sign}(\eta(x) - t^*_{D,\eta,\text{G-TP/PR}})$.

**Theorem 10** (**Consistency of Algorithm 1 w.r.t. G-TP/PR**). Let $\Psi = \Psi_{\text{G-TP/PR}}$. Let $D$ be a distribution on $\mathcal{X} \times \{\pm 1\}$ that satisfies Assumption A w.r.t. $\Psi_{\text{G-TP/PR}}$. If the CPE algorithm $\mathcal{C}$ in Algorithm 1 is $L_1$-consistent, then Algorithm 1 is G-TP/PR-consistent w.r.t. $D$.

*Proof.* We apply Theorem 1 noting that $\Psi_{\text{G-TP/PR}}(u,v,p) = \sqrt{\frac{pu^2}{pu+(1-p)(1-v)}}$ is continuous in each argument, and that (by Lemma 9) the Bayes optimal classifier for $\mathcal{P}^{\text{G-TP/PR}}$ is of the requisite form. $\qquad \square$

# 5  Consistency of Plug-in Algorithms for Non-decomposable Performance Measures that are Monotonic in TPR and TNR

The consistency results seen so far apply to any distribution that satisfies a mild continuity condition at the optimal threshold for a performance measure, and have crucially relied on the specific functional form of the measure. In this section, we shall see that under a stricter continuity assumption on the distribution, the empirical plug-in algorithm can be shown to be consistent w.r.t. any performance measure that is a continuous and monotonically increasing function of TPR and TNR.

**Assumption B.** We will say that a probability distribution $D$ on $\mathcal{X} \times \{\pm 1\}$ satisfies Assumption B w.r.t. $\Psi$ if $t^*_{D,\eta,\Psi}$ exists and is in $(0,1)$, and the cumulative distribution function of the random variable $\eta(x)$, $\mathbf{P}(\eta(x) \le z)$, is continuous at *all* $z \in (0,1)$.

Distributions that satisfy the above assumption also satisfy Assumption A. We show that under this assumption, the optimal classifier for any performance measure that is monotonically increasing in TPR and TNR is obtained by thresholding $\eta$, and this holds irrespective of the specific functional form of the measure. An application of Theorem 1 then gives us the desired consistency result.

**Lemma 11** (**Optimality of threshold classifiers for monotonic $\Psi$ under distributional assumption**). Let $\Psi : [0,1]^3 \to \mathbb{R}_+$ be monotonically increasing in its first two arguments. Then for any distribution $D$ on $\mathcal{X} \times \{\pm 1\}$ that satisfies Assumption B, the Bayes optimal classifier for $\mathcal{P}^\Psi$ is of the form $h^{\Psi,*}(x) = \text{sign}(\eta(x) - t^*_{D,\eta,\Psi})$.

**Theorem 12** (**Consistency of Algorithm 1 for monotonic $\Psi$ under distributional assumption**). Let $\Psi : [0,1]^3 \to \mathbb{R}_+$ be continuous in each argument, and monotonically increasing in its first two arguments. Let $D$ be a distribution on $\mathcal{X} \times \{\pm 1\}$ that satisfies Assumption B. If the CPE algorithm $\mathcal{C}$ in Algorithm 1 is $L_1$-consistent, then Algorithm 1 is $\Psi$-consistent w.r.t. $D$.

*Proof.* We apply Theorem 1 by using the continuity assumption on $\Psi$, and noting that, by Lemma 11 and monotonicity of $\Psi$, the Bayes optimal classifier for $\mathcal{P}^\Psi$ is of the requisite form. $\qquad \square$

The above result applies to all performance measures listed in Table 1, and in particular, to the geometric, harmonic, and quadratic means of TPR and TNR [2–5], for which the Bayes optimal classifier need not be of the requisite thresholded form for a general distribution (see Appendix C).

# 6  Experiments

We performed two types of experiments. The first involved synthetic data, where we demonstrate diminishing regret of the plug-in method in Algorithm 1 with growing sample size for different performance measures; since the data is generated from a known distribution, exact calculation of regret is possible here. The second involved real data, where we show that the plug-in algorithm is competitive with the state-of-the-art SVM$^{\text{perf}}$ algorithm for non-decomposable measures (SVMPerf) [12]; we also include for comparison a plug-in method with a fixed threshold of $0.5$ (Plug-in (0-1)). We consider three performance measures here: $F_1$-measure, G-TP/PR and G-Mean (see Table 1).

**Synthetic data.** We generated data from a known distribution (class conditionals are multivariate Gaussians with mixing ratio $p$ and equal covariance matrices) for which the optimal classifier for

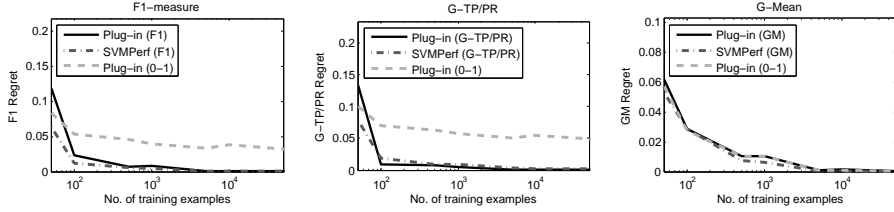

Figure 1: Experiments on synthetic data with $p = 0.5$: regret as a function of number of training examples using various methods for the $F_1$, G-TP/PR and G-mean performance measures.

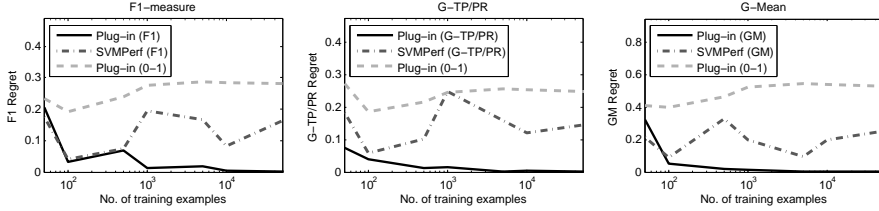

Figure 2: Experiments on synthetic data with $p = 0.1$: regret as a function of number of training examples using various methods for the $F_1$, G-TP/PR and G-Mean performance measures.

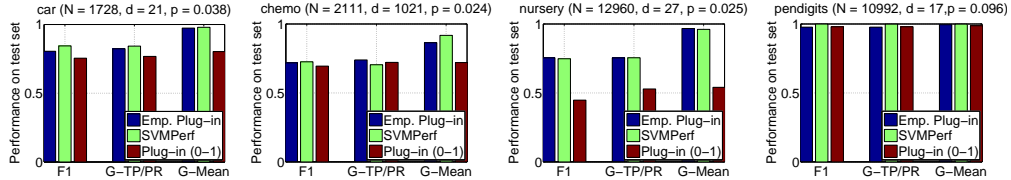

Figure 3: Experiments on real data: results for various methods (using linear models) on four data sets in terms of $F_1$, G-TP/PR and G-Mean performance measures. Here $N, d, p$ refer to the number of instances, number of features and fraction of positives in the data set respectively.

each performance measure considered here is linear, making it sufficient to learn a linear model; the distribution satisfies Assumption B w.r.t. each performance measure. We used regularized logistic regression as the CPE method in Algorithm 1 in order to satisfy the $L_1$-consistency condition in Theorem 1 (see Appendix A.1 and A.4 for details). The experimental results are shown in Figures 1 and 2 for $p = 0.5$ and $p = 0.1$ respectively. In each case, the regret for the empirical plug-in method (Plug-in (F1), Plug-in (G-TP/PR) and Plug-in (GM)) goes to zero with increasing training set size, validating our consistency results; SVM$^{\text{perf}}$ fails to exhibit diminishing regret for $p = 0.1$; and as expected, Plug-in (0-1), with its apriori fixed threshold, fails to be consistent in most cases.

**Real data.** We ran the three algorithms described earlier over data sets drawn from the UCI ML repository [29] and a cheminformatics data set obtained from [30], and report their performance on separately held test sets. Figure 3 contains results for four data sets averaged over 10 random train-test splits of the original data. (See Appendix A.2 for details and A.3 for additional results). Clearly, in most cases, the empirical plug-in method performs comparable to SVM$^{\text{perf}}$ and outperforms the Plug-in (0-1) method. Moreover, the empirical plug-in was found to run faster than SVM$^{\text{perf}}$.

# 7  Conclusions

We have presented a general method for proving consistency of plug-in algorithms that assign an empirical threshold to a suitable class probability estimate for a variety of non-decomposable performance measures for binary classification that can be expressed as a continuous function of TPR and TNR, and for which the Bayes optimal classifier is the class probability function thresholded suitably. We use our template to show consistency for the AM, $F_\beta$ and G-TP/PR measures, and under a continuous distribution, for any performance measure that is continuous and monotonic in TPR and TNR. Our experiments suggest that these algorithms are competitive with the SVM$^{\text{perf}}$ method.

### Acknowledgments

HN thanks support from a Google India PhD Fellowship. SA gratefully acknowledges support from DST, Indo-US Science and Technology Forum, and an unrestricted gift from Yahoo.

## Footnotes

[1]In the setting considered here, the goal is to maximize a (non-decomposable) function of expectations; we note that this is different from the decision-theoretic setting in [15], where one looks at the expectation of a non-decomposable performance measure on $n$ examples, and seeks to maximize its limiting value as $n \to \infty$.

[2]We say $\phi(S)$ converges in probability to $a \in \mathbb{R}$, written as $\phi(S) \xrightarrow{P} a$, if $\forall \epsilon > 0$, $\mathbf{P}_{S \sim D^n}(\vert \phi(S) - a \vert \geq \epsilon) \to 0$ as $n \to \infty$.

[3]For simplicity, we assume that $t^*_{D, \eta, \Psi}$ is in $(0, 1)$; our results easily extend to the case when $t^*_{D, \eta, \Psi} \in [0, 1]$.

[4]Note that the plug-in classification threshold chosen for the AM-measure is the same independent of the class probability estimate used; our consistency results will therefore apply in this case even if one uses, as in [16], the same sample for both learning a class probability estimate, and estimating the plug-in threshold.

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
