[Supplementary Material]

# On the Statistical Consistency of Plug-in Classifiers for Non-decomposable Performance Measures

## Appendix

## A  Experimental Details

### A.1  Synthetic data experiments

We use data drawn from a distribution $D$ over $(\mathcal{X} = \mathbb{R}^{10}) \times \{\pm 1\}$ that satisfies Assumption B (and therefore assumption A); recall that our plug-in consistency results for the $F_1$ and G-TP/PR measures apply to distributions that satisfy Assumption A (see Section 4), and the consistency result for G-Mean holds for distributions that satisfy Assumption B (see Section 5). The specifics of our experiments mirror those used in [16] and are listed here for completeness: positive examples ($y = 1$) are drawn from $\mathcal{N}(\mu, \Sigma)$ with probability $p \in (0, 1)$ and negative examples ($y = -1$) drawn from $\mathcal{N}(-\mu, \Sigma)$ with probability $(1 - p)$; $\mu$ is drawn uniformly at random from $\{\pm 1\}^{10}$ and $\Sigma \in \mathbb{R}^{10 \times 10}$ is drawn from a Wishart distribution with 20 degrees of freedom and a randomly drawn invertible positive semidefinite scale matrix. As pointed out earlier, the optimal classifier for each performance measure considered here under this distribution is linear, making it sufficient to learn a linear model (see Section A.4).

We evaluate the statistical regret of the empirical plug-in method (Algorithm 1 with $\alpha = 0.5$) and compare it against SVM$^{\text{perf}}$ with linear kernel (SVMPerf) adapted to optimize the performance measures considered here[5], and the Plug-in algorithm with a default threshold 0.5 (Plug-in (0-1)). The empirical plug-in algorithm (denoted for the three performance measures respectively as Plug-in (F1), Plug-in (G-TP/PR) and Plug-in (GM)) randomly splits the input data $S$ (drawn from $D$) into samples $S_1$ and $S_2$ for the purposes of learning a class probability estimate and choosing an appropriate threshold respectively; we use regularized (linear) logistic regression for learning a class probability estimate from $S_1$, with the regularization parameter set to $1/\sqrt{|S_1|}$, in order to satisfy the $L_1$-consistency requirement in Theorem 1 (see [16, 20] for details). The Plug-in (0-1) method learns a class probability estimate using the entire input data $(S_1, S_2)$, with the regularization parameter set to $1/\sqrt{|S|}$. The SVM$^{\text{perf}}$ algorithm also uses the entire input data $(S_1, S_2)$, with the regularization parameter selected from the range $\{10^{-3}, 10^{-2}, \ldots, 10^1\}$ via 5-fold cross-validation over the training sample[6].

### A.2  Real data experiments

For experiments with real data sets, we report the performance of the learned classifiers on separately held test data (we perform a random 2:1 train-test split of the original data, preserving class proportions). For the empirical plug-in algorithm, the parameter $\alpha$ in Algorithm 1 was set to 0.8. The regularization parameter for SVM$^{\text{perf}}$ was chosen from the range $\{10^{-7}, 10^{-2}, \ldots, 10^4\}$ and that for logistic regression was chosen from the range $\{10^{-3}, 10^{-2}, \ldots, 10^1\}$ using 5-fold cross validation over the corresponding training sample.

### A.3  Additional results on real data

Table 2 summarizes all the real data sets that have been used in our experiments (both in Section 6 and this section). Figure 4 shows the test performances of the plug-in (with both the empirically chosen threshold and default threshold) and SVM$^{\text{perf}}$ methods w.r.t. $F_1$, G-TP/PR and G-Mean measures over data sets included in Table 2 that were not already covered in Figure 3. Table 3 lists experiment outputs for all datasets, all algorithms and all performance measures. Once again, it

Figure 4: Experiments on real data: results for empirical Plug-in, SVM$^{\text{perf}}$ and Plug-in (0-1) methods (with linear models) on several UCI data sets in terms of $F_1$, G-TP/PR and G-Mean performance measures. Here $N, d, p$ refer to the number of instances, number of features and fraction of positives in the data set respectively.

can be observed that the empirical Plug-in is competitive with SVM$^{\text{perf}}$ and outperforms the Plug-in (0-1) method in most cases.

Table 2: Summary of real data sets used in this study

| Data sets | #examples | #features | $p = \mathbf{P}(y = 1)$ |
|---|---|---|---|
| car | 1728 | 21 | 0.038 |
| chemo-a1a | 2111 | 1021 | 0.024 |
| nursery | 12960 | 27 | 0.026 |
| pendigits | 10992 | 17 | 0.096 |
| letter | 18668 | 16 | 0.034 |
| optdigits | 5620 | 64 | 0.901 |
| segment | 2086 | 19 | 0.143 |
| spambase | 4210 | 57 | 0.398 |
| splice | 3005 | 240 | 0.226 |
| thyroid | 7129 | 21 | 0.977 |

## A.4 Regret calculation for synthetic data

As mentioned in Section A.1, the distribution $D$ over $\mathbb{R}^{10} \times \{\pm 1\}$ that we consider consists of multivariate Gaussian class conditional distributions, with positive instances being drawn from $\mathcal{N}(\mu, \Sigma)$ and negative instances being drawn from $\mathcal{N}(-\mu, \Sigma)$. We denote the probability density functions (pdfs) corresponding to $x|y = 1$ and $x|y = -1$ as $f_+$ and $f_-$ respectively.

We first show that any classifier obtained by thresholding the the class probability function $\eta$ under the above distribution is linear. For any $x \in \mathbb{R}^{10}$, we have

$$\eta(x) = \mathbf{P}(y = 1|x) = \frac{\mathbf{P}(x|y = 1).\mathbf{P}(y = 1)}{\mathbf{P}(x|y = 1).\mathbf{P}(y = 1) + \mathbf{P}(x|y = -1).\mathbf{P}(y = -1)}$$

$$= \frac{p.f_+(x)}{p.f_+(x) + (1 - p)f_-(x)}$$

$$= \frac{1}{1 + e^{-f(x)}},$$

where $f(x) = \ln\left(\frac{p.f_+(x)}{(1-p)f_-(x)}\right) = 2\mu^T \Sigma^{-1} x + \ln\left(\frac{p}{1-p}\right)$ turns out to be a linear function of $x$. As a result, any thresholded classifier of the form $\text{sign} \circ (\eta(x) - c)$, for some $c \in (0, 1)$, can be written as a linear classifier: $\text{sign} \circ (f(x) - \ln(c/(1 - c)))$.

Table 3: Results from experiments on all real datasets. For each dataset and performance measure, the algorithm outputs with the highest and second-highest mean performance are highlighted in boldface and italics respectively.

| Data sets | Algorithm | $F_1$ | G-TP/PR | G-Mean |
|---|---|---|---|---|
| car | Emp. Plug-in | *0.8053 ± 0.0777* | *0.8231 ± 0.0691* | *0.9738 ± 0.0164* |
| | SVM-Perf | **0.8442 ± 0.0327** | **0.8426 ± 0.0367** | **0.9797 ± 0.0179** |
| | Plug-in (0-1) | 0.7550 ± 0.0643 | 0.7679 ± 0.0598 | 0.8027 ± 0.0517 |
| chemo-a1a | Emp. Plug-in | *0.7182 ± 0.1286* | **0.7393 ± 0.1044** | *0.8640 ± 0.0997* |
| | SVM$^{\text{perf}}$ | **0.7256 ± 0.0541** | *0.7044 ± 0.0560* | **0.9172 ± 0.0801** |
| | Plug-in (0-1) | 0.6945 ± 0.0659 | 0.7214 ± 0.0557 | 0.7202 ± 0.0382 |
| nursery | Emp. Plug-in | **0.7545 ± 0.0319** | *0.7542 ± 0.0302* | **0.9668 ± 0.0162** |
| | SVM$^{\text{perf}}$ | *0.7479 ± 0.0138* | **0.7549 ± 0.0120** | *0.9609 ± 0.0084* |
| | Plug-in (0-1) | 0.4478 ± 0.0425 | 0.5287 ± 0.0335 | 0.5406 ± 0.0334 |
| pendigits | Emp. Plug-in | 0.9770 ± 0.0073 | 0.9772 ± 0.0072 | *0.9944 ± 0.0037* |
| | SVM$^{\text{perf}}$ | **0.9991 ± 0.0009** | **0.9984 ± 0.0018** | **0.9986 ± 0.0010** |
| | Plug-in (0-1) | *0.9795 ± 0.0063* | *0.9795 ± 0.0063* | 0.9873 ± 0.0068 |
| letter | Emp. Plug-in | *0.6685 ± 0.0239* | *0.6668 ± 0.0240* | **0.9093 ± 0.0104** |
| | SVM$^{\text{perf}}$ | **0.7143 ± 0.0324** | **0.7144 ± 0.0304** | *0.9017 ± 0.0138* |
| | Plug-in (0-1) | 0.0876 ± 0.0268 | 0.2117 ± 0.0359 | 0.2117 ± 0.0359 |
| optdigits | Emp. Plug-in | *0.9986 ± 0.0005* | *0.9912 ± 0.0034* | *0.9986 ± 0.0005* |
| | SVM$^{\text{perf}}$ | 0.9985 ± 0.0005 | **0.9925 ± 0.0020** | 0.9985 ± 0.0006 |
| | Plug-in (0-1) | **0.9987 ± 0.0002** | 0.9888 ± 0.0023 | **0.9987 ± 0.0002** |
| segment | Emp. Plug-in | 0.9911 ± 0.0114 | 0.9911 ± 0.0113 | 0.9899 ± 0.0082 |
| | SVM$^{\text{perf}}$ | **0.9964 ± 0.0034** | **0.9964 ± 0.0033** | **0.9961 ± 0.0034** |
| | Plug-in (0-1) | *0.9959 ± 0.0031* | *0.9959 ± 0.0031* | *0.9959 ± 0.0031* |
| spambase | Emp. Plug-in | *0.8489 ± 0.0145* | *0.8749 ± 0.0101* | *0.8493 ± 0.0143* |
| | SVM$^{\text{perf}}$ | **0.9078 ± 0.0082** | **0.9250 ± 0.0068** | **0.9077 ± 0.0088** |
| | Plug-in (0-1) | 0.8076 ± 0.0129 | 0.8317 ± 0.0105 | 0.8126 ± 0.0126 |
| splice | Emp. Plug-in | *0.9391 ± 0.0100* | *0.9393 ± 0.0099* | *0.9615 ± 0.0097* |
| | SVM$^{\text{perf}}$ | 0.9264 ± 0.0133 | 0.9268 ± 0.0142 | 0.9570 ± 0.0060 |
| | Plug-in (0-1) | **0.9465 ± 0.0093** | **0.9466 ± 0.0093** | **0.9636 ± 0.0058** |
| thyroid | Emp. Plug-in | *0.9941 ± 0.0008* | *0.9941 ± 0.0008* | *0.9293 ± 0.0240* |
| | SVM$^{\text{perf}}$ | **0.9950 ± 0.0009** | **0.9952 ± 0.0008** | **0.9784 ± 0.0100** |
| | Plug-in (0-1) | 0.9887 ± 0.0002 | 0.9887 ± 0.0002 | 0.1769 ± 0.0749 |

We next describe how one can compute the $\Psi$-regret of a linear classifier $h : \mathbb{R}^{10} \to \{\pm 1\}$ under the given distribution:

$$\text{regret}_D^{\Psi}[h] = \mathcal{P}_D^{\Psi,*} - \mathcal{P}_D^{\Psi}[h].$$

In particular, we shall describe how the values of $\mathcal{P}_D^{\Psi}[h]$ and $\mathcal{P}_D^{\Psi,*}$ in the above expression can be computed for the given distribution $D$.

We start with the procedure outlined in [16] for calculating the performance measure $\mathcal{P}_D^{\Psi}$ for any linear classifier $h(x) = \text{sign} \circ (w^{\top}x + b)$, where (for our purpose) $w \in \mathbb{R}^{10}$ and $b \in \mathbb{R}$. The TPR of $h$ is given by:

$$\text{TPR}_D[h] = \mathbf{P}(h(x) = 1 | y = 1) = \int_{x \,|\, w^{\top}x + b \geq 0} f_+(x)\,dx.$$

It can be seen that $w^{\top}x \,|\, y = 1$ follows the normal distribution $\mathcal{N}(w^{\top}x,\, w^{\top}\Sigma w)$, and therefore by change of variables, we have

$$\text{TPR}_D[h] = \int_{-b}^{\infty} g_+(x)\,dx,$$

where $g_+$ is the pdf corresponding to $\mathcal{N}(w^\top x, w^\top \Sigma w)$. Likewise, the TNR for $h$ is given by:

$$\text{TNR}_D[h] = \int_{-\infty}^{-b} g_-(x)\, dx,$$

where $g_-$ is the pdf corresponding to $\mathcal{N}(-w^\top x,\ w^\top \Sigma w)$. This way, given $w$ and $b$, both TPR and TNR are straightforward to determine, and consequently so is any performance measure that is a function $\Psi$ of these quantities.

We next describe how the optimal value of the given performance measure $\mathcal{P}_D^{\Psi,*}$ can be computed. Since the given distribution satisfies Assumptions A and B, the optimal classifier for all performance measures considered in this work is obtained by suitably thresholding the class probability function $\eta$; hence the optimal value $\mathcal{P}_D^{\Psi,*}$ for the given measure can be computed by performing a line search over $(0,1)$ and picking the threshold $c^* \in (0,1)$ for which the linear classifier $\text{sign} \circ \big(f(x) - c^*\big)$ maximizes the performance measure.

## B  Complete Proofs for Lemmas

### B.1  Complete proof for Lemma 2

*Proof.* First, we simplify what we need to prove. We need to show that for a fixed $c \in (0,1)$,

$$\text{TPR}_D[\text{sign} \circ (\widehat{\eta}_S(x) - c)] \xrightarrow{P} \text{TPR}_D[\text{sign} \circ (\eta(x) - c)] \text{ and}$$

$$\text{TNR}_D[\text{sign} \circ (\widehat{\eta}_S(x) - c)] \xrightarrow{P} \text{TNR}_D[\text{sign} \circ (\eta(x) - c)]$$

$$\Longleftrightarrow \quad \mathbf{P}\big(\text{sign} \circ (\widehat{\eta}_S(x) - c) = 1 \,|\, y = 1\big) \xrightarrow{P} \mathbf{P}\big(\text{sign} \circ (\eta(x) - c) = 1 \,|\, y = 1\big) \text{ and}$$

$$\mathbf{P}\big(\text{sign} \circ (\widehat{\eta}_S(x) - c) = -1 \,|\, y = -1\big) \xrightarrow{P} \mathbf{P}\big(\text{sign} \circ (\eta(x) - c) = -1 \,|\, y = -1\big)$$

$$\Longleftrightarrow \quad \mathbf{P}\big(\widehat{\eta}_S(x) > c \,|\, y = 1\big) \xrightarrow{P} \mathbf{P}\big(\eta(x) > c \,|\, y = 1\big) \text{ and}$$

$$\mathbf{P}\big(\widehat{\eta}_S(x) \le c \,|\, y = -1\big) \xrightarrow{P} \mathbf{P}\big(\eta(x) \le c \,|\, y = -1\big)$$

$$\Longleftrightarrow \quad \mathbf{P}\big(\widehat{\eta}_S(x) \le c \,|\, y = 1\big) \xrightarrow{P} \mathbf{P}\big(\eta(x) \le c \,|\, y = 1\big) \text{ and}$$

$$\mathbf{P}\big(\widehat{\eta}_S(x) \le c \,|\, y = -1\big) \xrightarrow{P} \mathbf{P}\big(\eta(x) \le c \,|\, y = -1\big)$$

$$\Longleftrightarrow \quad \mathbf{P}_{x|y=1}\big(\widehat{\eta}_S(x) \le c\big) \xrightarrow{P} \mathbf{P}_{x|y=1}\big(\eta(x) \le c\big) \text{ and}$$

$$\mathbf{P}_{x|y=-1}\big(\widehat{\eta}_S(x) \le c\big) \xrightarrow{P} \mathbf{P}_{x|y=-1}\big(\eta(x) \le c\big). \tag{3}$$

We now analyze the $L_1$-consistency guarantee assumed in the statement of Lemma 2, namely $\mathbf{E}_x\big[|\widehat{\eta}_S(x) - \eta(x)|\big] \xrightarrow{P} 0$. We begin by expanding this term.

$$\mathbf{E}_x\big[|\widehat{\eta}_S(x) - \eta(x)|\big] = p.\mathbf{E}_x\big[|\widehat{\eta}_S(x) - \eta(x)| \,|\, y = 1\big] + (1-p).\mathbf{E}_x\big[|\widehat{\eta}_S(x) - \eta(x)| \,|\, y = -1\big]$$

$$= p.\mathbf{E}_{x|y=1}\big[|\widehat{\eta}_S(x) - \eta(x)|\big] + (1-p).\mathbf{E}_{x|y=-1}\big[|\widehat{\eta}_S(x) - \eta(x)|\big].$$

Using the above expansion and the given guarantee on $\widehat{\eta}_S$ (along with $p \in (0,1)$), we obtain $\mathbf{E}_{x|y=1}\big[|\widehat{\eta}_S(x) - \eta(x)|\big] \xrightarrow{P} 0$ and $\mathbf{E}_{x|y=-1}\big[|\widehat{\eta}_S(x) - \eta(x)|\big] \xrightarrow{P} 0$[7]. Applying Markov inequality for the random variable $|\widehat{\eta}_S(x) - \eta(x)|$ for a fixed $S$, we have for any $\epsilon_1 > 0$,

$$\mathbf{P}_{x|y=1}\big(|\widehat{\eta}_S(x) - \eta(x)| \ge \epsilon_1\big) \le \frac{\mathbf{E}_{x|y=1}\big[|\widehat{\eta}_S(x) - \eta(x)|\big]}{\epsilon_1}$$

$$\text{and} \quad \mathbf{P}_{x|y=-1}\big(|\widehat{\eta}_S(x) - \eta(x)| \ge \epsilon_1\big) \le \frac{\mathbf{E}_{x|y=-1}\big[|\widehat{\eta}_S(x) - \eta(x)|\big]}{\epsilon_1},$$

which in turn yields for a fixed $\epsilon_1 > 0$,

$$\mathbf{P}_{x|y=1}\big(|\widehat{\eta}_S(x) - \eta(x)| \ge \epsilon_1\big) \xrightarrow{P} 0; \tag{4}$$

$$\mathbf{P}_{x|y=-1}\big(|\widehat{\eta}_S(x) - \eta(x)| \geq \epsilon_1\big) \xrightarrow{P} 0, \tag{5}$$

where recall that the convergence in probability is w.r.t. to a random draw of $S$ according to $D^n$.

In the rest of the proof, we shall make use of (a) the fact that Eq. (4) and (5) hold for arbitrarily small values of $\epsilon_1$ and (b) our assumption that $\mathbf{P}(\eta(x) \geq c \,|\, y = 1)$ and $\mathbf{P}(\eta(x) \geq c \,|\, y = -1)$ are continuous at $c \in (0, 1)$ to establish the desired result. We start proving the result w.r.t. $x|y = 1$. For a fixed $S$ and a fixed $\epsilon_2 > 0$, we have

$$\begin{aligned}
&\mathbf{P}_{x|y=1}\big(\widehat{\eta}_S(x) \leq c\big) \\
&= \mathbf{P}_{x|y=1}\big(\widehat{\eta}_S(x) \leq c,\, \eta(x) \leq c + \epsilon_2\big) + \mathbf{P}_{x|y=1}\big(\widehat{\eta}_S(x) \leq c, \eta(x) > c + \epsilon_2\big) \\
&\leq \mathbf{P}_{x|y=1}\big(\eta(x) \leq c + \epsilon_2\big) + \mathbf{P}_{x|y=1}\big(|\widehat{\eta}_S(x) - \eta(x)| \geq \epsilon_2\big),
\end{aligned} \tag{6}$$

and

$$\begin{aligned}
&\mathbf{P}_{x|y=1}\big(\eta(x) \leq c - \epsilon_2\big) \\
&= \mathbf{P}_{x|y=1}\big(\widehat{\eta}_S(x) \leq c,\, \eta(x) \leq c - \epsilon_2\big) + \mathbf{P}_{x|y=1}\big(\widehat{\eta}_S(x) > c, \eta(x) \leq c - \epsilon_2\big) \\
&\leq \mathbf{P}_{x|y=1}\big(\widehat{\eta}_S(x) \leq c\big) + \mathbf{P}_{x|y=1}\big(|\widehat{\eta}_S(x) - \eta(x)| \geq \epsilon_2\big).
\end{aligned} \tag{7}$$

Consequently from Eq. (6) and (7), we get

$$\mathbf{P}_{x|y=1}\big(\eta(x) \leq c - \epsilon_2\big) - \mathbf{P}_{x|y=1}\big(|\widehat{\eta}_S(x) - \eta(x)| \geq \epsilon_2\big) \leq \mathbf{P}_{x|y=1}\big(\widehat{\eta}_S(x) \leq c\big)$$
$$\text{and } \mathbf{P}_{x|y=1}\big(\widehat{\eta}_S(x) \leq c\big) \leq \mathbf{P}_{x|y=1}\big(\eta(x) \leq c + \epsilon_2\big) + \mathbf{P}_{x|y=1}\big(|\widehat{\eta}_S(x) - \eta(x)| \geq \epsilon_2\big).$$

Subtracting the term $\mathbf{P}_{x|y=1}\big(\eta(x) \leq c\big)$ from both sides in each of the above inequalities and combining the resulting inequalities then gives us

$$\begin{aligned}
&\big|\mathbf{P}_{x|y=1}\big(\widehat{\eta}_S(x) \leq c\big) - \mathbf{P}_{x|y=1}\big(\eta(x) \leq c\big)\big| \leq \\
&\max \Bigg\{ \underbrace{\mathbf{P}_{x|y=1}\big(|\widehat{\eta}_S(x) - \eta(x)| \geq \epsilon_2\big) + \mathbf{P}_{x|y=1}\big(\eta(x) \leq c + \epsilon_2\big) - \mathbf{P}_{x|y=1}\big(\eta(x) \leq c\big)}_{\text{term}_1}, \\
&\quad \underbrace{\mathbf{P}_{x|y=1}\big(|\widehat{\eta}_S(x) - \eta(x)| \geq \epsilon_2\big) - \mathbf{P}_{x|y=1}\big(\eta(x) \leq c - \epsilon_2\big) + \mathbf{P}_{x|y=1}\big(\eta(x) \leq c\big)}_{\text{term}_2} \Bigg\}.
\end{aligned} \tag{8}$$

Keeping $S$ fixed, we now allow $\epsilon_2 \to 0$. In particular, by our assumption that $\mathbf{P}_{x|y=1}\big(\eta(x) \leq c\big)$ is continuous at $c$, for the terms inside the above 'max', we have:

$$\lim_{\epsilon_2 \to 0} \text{term}_1 = \lim_{\epsilon_2 \to 0} \mathbf{P}_{x|y=1}\big(|\widehat{\eta}_S(x) - \eta(x)| \geq \epsilon_2\big);$$

$$\lim_{\epsilon_2 \to 0} \text{term}_2 = \lim_{\epsilon_2 \to 0} \mathbf{P}_{x|y=1}\big(|\widehat{\eta}_S(x) - \eta(x)| \geq \epsilon_2\big).$$

Thus for a fixed $S$, the following holds from Eq. (8):

$$0 \leq \big|\mathbf{P}_{x|y=1}\big(\widehat{\eta}_S(x) \leq c\big) - \mathbf{P}_{x|y=1}\big(\eta(x) \leq c\big)\big| \leq \lim_{\epsilon_2 \to 0} \mathbf{P}_{x|y=1}\big(|\widehat{\eta}_S(x) - \eta(x)| \geq \epsilon_2\big).$$

Now, from an application of Eq. (4) (which holds for arbitrarily small $\epsilon_1$), we obtain the following convergence in probability over a random draw of $S$ from $D^n$:

$$\big|\mathbf{P}_{x|y=1}\big(\widehat{\eta}_S(x) \leq c\big) - \mathbf{P}_{x|y=1}\big(\eta(x) \leq c\big)\big| \xrightarrow{P} 0,$$

which in turn, implies

$$\mathbf{P}_{x|y=1}\big(\widehat{\eta}_S(x) \leq c\big) \xrightarrow{P} \mathbf{P}_{x|y=1}\big(\eta(x) \leq c\big).$$

This is the desired relation w.r.t $x|y = 1$ (as seen in Eq. (3)). The desired result w.r.t. $x|y = -1$ follows likewise. $\qquad\square$

## B.2 Complete proof for Lemma 4

*Proof.* Define for $i, j \in \{-1, 1\}$:

$$\widehat{p}_{i,j,n}[h] = \sum_{k=1}^{n} \mathbf{1}(y_k = i,\ h(x_k) = j)/n \quad \text{and} \quad p_{ij}[h] = \mathbf{E}_D\big[\mathbf{1}(y = i,\ h(x) = j)\big].$$

For a fixed $h \in \mathcal{T}_f$, by the weak law of large numbers (WLLN), we have $\forall i, j$:

$$\widehat{p}_{i,j,n}[h] \xrightarrow{P} p_{i,j}[h],$$

where the convergence in probability is over draw of $S \sim D^n$. Also, $\widehat{p}_S \xrightarrow{P} p$ (again by WLLN). Given that

$$\widehat{\text{TPR}}_S[h] = \frac{1}{\widehat{p}_S}\widehat{p}_{1,1,n}[h] \quad \text{and} \quad \widehat{\text{TNR}}_S[h] = \frac{1}{(1 - \widehat{p}_S)}\widehat{p}_{-1,-1,n}[h],$$

we thus have

$$\widehat{\text{TPR}}_S[h] \xrightarrow{P} \frac{p_{1,1}[h]}{p} = \text{TPR}_D[h] \quad \text{and} \quad \widehat{\text{TNR}}_S[h] \xrightarrow{P} \frac{p_{-1,-1}[h]}{1 - p} = \text{TNR}_D[h].$$

In turn, by continuity of $\Psi$, we obtain

$$\widehat{\mathcal{P}}_S^\Psi[h] \xrightarrow{P} \mathcal{P}_D^\Psi[h].$$

$\square$

## B.3 Complete proof for Lemma 5

*Proof.* Recall that any fixed $h \in \mathcal{T}_f$ is of the form $\text{sign} \circ (f(x) - c)$ for some constant $c \in (0, 1)$. Let $p_{i,j}[h]$ and $p_{i,j,n}[h]$ be as defined in the proof of Lemma 4 (Section B.2). Since VC-dimension$(\mathcal{T}_f) = 1$, by standard VC-dimension based uniform convergence arguments, we can argue that for all $i, j \in \{\pm 1\}$, given any $\epsilon' > 0$,

$$\mathbf{P}_{S \sim D^n}\left(\bigcup_{h \in \mathcal{T}_f} \big|\widehat{p}_{i,j,n}[h] - p_{i,j}[h]\big| \geq \epsilon'\right) \to 0.$$

We also have $\widehat{p}_S \xrightarrow{P} p$ (by WLLN).

Now, as in the proof of Lemma 4, observing that TPR and TNR are continuous functions of the above quantities, it can be shown using an appropriate choice of $\epsilon' > 0$ in the above expressions, and by an application of union bound, that for any given $\epsilon > 0$,

$$\mathbf{P}_{S \sim D^n}\left(\bigcup_{h \in \mathcal{T}_f} \left\{\big|\text{TPR}_D[h] - \widehat{\text{TPR}}_S[h]\big| \geq \epsilon\right\}\right) \to 0$$

$$\text{and} \quad \mathbf{P}_{S \sim D^n}\left(\bigcup_{h \in \mathcal{T}_f} \left\{\big|\text{TNR}_D[h] - \widehat{\text{TNR}}_S[h]\big| \geq \epsilon\right\}\right) \to 0.$$

Once again, by continuity of $\Psi$, we have:

$$\mathbf{P}_{S \sim D^n}\left(\bigcup_{h \in \mathcal{T}_f} \left\{\big|\mathcal{P}_D^\Psi[h] - \widehat{\mathcal{P}}_S^\Psi[h]\big| \geq \epsilon\right\}\right) \to 0 \text{ as } n \to \infty,$$

as desired. $\square$

## B.4 Complete proof for Lemma 9

*Proof.* Our proof is similar to that of Theorem 4 in [15]. Recall $\mathcal{T}_\eta = \{\text{sign} \circ (\eta - t) \,|\, t \in [0, 1]\}$ and let $h^* = \sup_{h \in \mathcal{T}_\eta} \mathcal{P}_D^{\text{G-TP/PR}}[h]$ (the existence of this classifier is guaranteed by Assumption A). We shall show that for any $h \notin \mathcal{T}_\eta$, $\exists \widetilde{h} \in \mathcal{T}_\eta$ such that $\mathcal{P}_D^{\text{G-TP/PR}}[\widetilde{h}] \geq \mathcal{P}_D^{\text{G-TP/PR}}[h]$, thus giving us $\mathcal{P}_D^{\text{G-TP/PR}}[h^*] \geq \mathcal{P}_D^{\text{G-TP/PR}}[\widetilde{h}] \geq \mathcal{P}_D^{\text{G-TP/PR}}[h]$; this would imply that the optimal predictor for G-TP/PR is indeed of the desired threshold form.

For any $h \notin \mathcal{T}_\eta$, upon arranging all instances $x \in \mathcal{X}$ in non-increasing order of $\eta$, we can find disjoint subsets $A, B, C \subseteq \mathcal{X}$, with $\sup_{x \in A} \eta(x) \leq \inf_{x \in B} \eta(x) \leq \sup_{x \in B} \eta(x) \leq \inf_{x \in C} \eta(x)$, such that: $A \cup C = \{x \in \mathcal{X} \,|\, h(x) = 1\}$ and $B \subseteq \{x \in \mathcal{X} \,|\, h(x) = -1\}$. We now define two new classifiers:

$$h_A(x) = \begin{cases} -1 & \text{if } x \in A \\ h(x) & \text{o/w} \end{cases} \quad \text{and} \quad h_B(x) = \begin{cases} 1 & \text{if } x \in B \\ h(x) & \text{o/w} \end{cases}.$$

We now claim that one of these newly defined classifiers must be at least as good as $h$ w.r.t. G-TP/PR (this claim is proved below).

*Claim.* Either $\mathcal{P}_D^{\text{G-TP/PR}}[h_A] \geq \mathcal{P}_D^{\text{G-TP/PR}}[h]$ or $\mathcal{P}_D^{\text{G-TP/PR}}[h_B] \geq \mathcal{P}_D^{\text{G-TP/PR}}[h]$.

According to the above claim, any classifier that is not of the form $\text{sign} \circ (\eta(x) - c)$ is only as good as one of $h_B$ or $h_A$ w.r.t. G-TP/PR. We could now imagine one of $h_A$ or $h_B$ as the new $h$ and make repeated use of the above exchange argument to eventually arrive at a classifier $\widetilde{h}$ in $\mathcal{T}_\eta$ with $\mathcal{P}_D^{\text{G-TP/PR}}[\widetilde{h}] \geq \mathcal{P}_D^{\text{G-TP/PR}}[h]$, as desired.

It remains to be shown that the above claim is true.

*Proof of Claim.* Let us assume the contrary, that $\mathcal{P}_D^{\text{G-TP/PR}}[h] > \mathcal{P}_D^{\text{G-TP/PR}}[h_A]$ and $\mathcal{P}_D^{\text{G-TP/PR}}[h] > \mathcal{P}_D^{\text{G-TP/PR}}[h_B]$, and arrive at a contradiction. Let $a = \mathbf{P}(x \in A)$, $b = \mathbf{P}(x \in B)$ and $c = \mathbf{P}(x \in C)$, and assume without loss of generality that $a, b > 0$. Let $\alpha = \mathbf{E}_x\big[\eta(x) \,\big|\, x \in A\big]$, $\beta = \mathbf{E}_x\big[\eta(x) \,\big|\, x \in B\big]$ and $\gamma = \mathbf{E}_x\big[\eta(x) \,\big|\, x \in C\big]$. It is clearly seen that $0 \leq \alpha \leq \beta \leq \gamma$. With the above definitions, we have, $\text{TPR}_D(h) = \frac{a\alpha + c\gamma}{p}$ and $\text{Prec}_D(h) = \frac{a\alpha + c\gamma}{a + c}$, and in turn, $\mathcal{P}^{\text{G-TP/PR}}(h) = \sqrt{\frac{(a\alpha + c\gamma)^2}{p(a+c)}}$, while $\mathcal{P}^{\text{G-TP/PR}}(h_B) = \sqrt{\frac{(a\alpha + b\beta + c\gamma)^2}{p(a+b+c)}}$ and $\mathcal{P}^{\text{G-TP/PR}}(h_A) = \sqrt{\frac{(c\gamma)^2}{p(c)}}$.

By our contradiction hypothesis,

$$\sqrt{\frac{(a\alpha + c\gamma)^2}{p(a+c)}} > \sqrt{\frac{(a\alpha + b\beta + c\gamma)^2}{p(a+b+c)}} \quad \text{and} \quad \sqrt{\frac{(a\alpha + c\gamma)^2}{p(a+c)}} > \sqrt{\frac{(c\gamma)^2}{p(c)}},$$

which implies

$$(a + b + c)(a\alpha + c\gamma)^2 > (a + c)(a\alpha + b\beta + c\gamma)^2 \tag{9}$$
$$\text{and} \quad c(a\alpha + c\gamma)^2 > (a + c)(c\gamma)^2. \tag{10}$$

Now, from Eq. (10), we have

$$c(a^2\alpha^2 + c^2\gamma^2 + 2a\alpha c\gamma) > ac^2\gamma^2 + c^3\gamma^2 \quad \text{or} \quad ac\alpha^2 + 2c^2\alpha\gamma > c^2\gamma^2, \tag{11}$$

Next, from Eq. (9), we have

$$(a + b + c)(a^2\alpha^2 + c^2\gamma^2 + 2ac\alpha\gamma) > (a + c)(a^2\alpha^2 + b^2\beta^2 + c^2\gamma^2 + 2ab\alpha\beta + 2bc\beta\gamma + 2ac\alpha\gamma),$$

which can be simplified as

$$b(a^2\alpha^2 + c^2\gamma^2 + 2ac\alpha\gamma) > (a + c)(b^2\beta^2 + 2ab\alpha\beta + 2bc\beta\gamma).$$

Using the upper bound for the term $c^2\gamma^2$ from Eq. (11) in the above inequality, we get

$$b(a^2\alpha^2 + ac\alpha^2 + 2c^2\alpha\gamma + 2ac\alpha\gamma) > (a + c)(b^2\beta^2 + 2ab\alpha\beta + 2bc\beta\gamma)$$
$$\implies \quad b(a + c)(a\alpha^2 + 2c\alpha\gamma) > (a + c)(b^2\beta^2 + 2ab\alpha\beta + 2bc\beta\gamma)$$

$$\implies \quad a\alpha^2 + 2c\alpha\gamma \;>\; b\beta^2 + 2a\alpha\beta + 2c\beta\gamma.$$

Using $\beta \geq \alpha$, we can now lower bound the right hand side in the above inequality to get

$$a\alpha^2 + 2c\alpha\gamma \;>\; b\beta^2 + 2a\alpha^2 + 2c\alpha\gamma \quad \implies \quad 0 \;>\; a\alpha^2 + b\beta^2,$$

which is a contradiction since $a, b > 0$ and $\alpha, \beta \geq 0$. This proves the claim. $\qquad\square$

## B.5 Complete proof for Lemma 11

*Proof.* Recall $\mathcal{T}_\eta = \{\mathrm{sign} \circ (\eta - t) \,|\, t \in [0,1]\}$ and let $h^* = \sup_{h \in \mathcal{T}_\eta} \mathcal{P}_D^\Psi[h]$ (the existence of this classifier is guaranteed by Assumption B). We shall now use an exchange argument (that makes use of Assumption B) to show that for any $h \notin \mathcal{T}_\eta$, $\exists\, \widetilde{h} \in \mathcal{T}_\eta$ such that $\mathcal{P}_D^\Psi[\widetilde{h}] \geq \mathcal{P}_D^\Psi[h]$, thus implying $\mathcal{P}_D^\Psi[h^*] \geq \mathcal{P}_D^\Psi[\widetilde{h}] \geq \mathcal{P}_D^\Psi[h]$; this would imply that the optimal predictor for $\mathcal{P}^\Psi$ is indeed of the desired threshold form. In particular, we shall show that $\mathrm{TPR}_D[\widetilde{h}] \geq \mathrm{TPR}_D[h]$ and $\mathrm{TNR}_D[\widetilde{h}] \geq \mathrm{TNR}_D[h]$, which by the monotonicity assumption on $\Psi$ yields $\mathcal{P}_D^\Psi[\widetilde{h}] \geq \mathcal{P}_D^\Psi[h]$.

For any $h \notin \mathcal{T}_\eta$, upon arranging all instances $x \in \mathcal{X}$ in non-increasing order of $\eta$, we can find disjoint subsets $A \subseteq \{x \in \mathcal{X} \,|\, h(x) = 1\}$ and $B \subseteq \{x \in \mathcal{X} \,|\, h(x) = -1\}$ such that $\sup_{x \in A} \eta(x) \leq \inf_{x \in B} \eta(x)$. Let $a = \mathbf{P}(x \in A)$ and $b = \mathbf{P}(x \in B)$; assume without loss of generality, $a, b > 0$.

Let us consider the case where $a \geq b$; here we choose a set $A' \subseteq A$ with $\mathbf{P}(x \in A') = b$, and define a classifier $h'$ as

$$h'(x) = \begin{cases} 1 & \text{if } x \in B \\ -1 & \text{if } x \in A' \ . \\ h(x) & \text{o/w} \end{cases}$$

We shall now show that $\mathrm{TPR}_D[h'] \geq \mathrm{TPR}_D[h]$ and $\mathrm{TNR}_D[h'] \geq \mathrm{TNR}_D[h]$. In particular,

$$
\begin{aligned}
& \mathrm{TPR}_D[h'] - \mathrm{TPR}_D[h] \\
&= \mathbf{P}\big(h'(x) = 1 \,|\, y = 1\big) - \mathbf{P}\big(h(x) = 1 \,|\, y = 1\big) \\
&= \frac{1}{p}\mathbf{E}_x\big[\eta(x)\mathbf{1}\big(h'(x) = 1\big)\big] - \frac{1}{p}\mathbf{E}_x\big[\eta(x)\mathbf{1}\big(h(x) = 1\big)\big] \\
&= \frac{1}{p}\Big[\mathbf{E}_x\big[\eta(x)\mathbf{1}\big(h'(x) = 1,\, h(x) = -1\big)\big] - \mathbf{E}_x\big[\eta(x)\mathbf{1}\big(h(x) = 1,\, h'(x) = -1\big)\big]\Big] \\
&= \frac{1}{p}\Big[\mathbf{E}_x\big[\eta(x)\mathbf{1}\big(x \in B\big)\big] - \mathbf{E}_x\big[\eta(x)\mathbf{1}\big(x \in A'\big)\big]\Big] \quad \text{(by definition of } h') \\
&\geq \frac{1}{p}\Big[\big(\inf_{x \in B} \eta(x)\big)\mathbf{P}(x \in B) - \big(\sup_{x \in A'} \eta(x)\big)\mathbf{P}(x \in A')\Big] \\
&= \frac{b}{p}\Big[\inf_{x \in B} \eta(x) - \sup_{x \in A'} \eta(x)\Big] \quad \text{(using } \mathbf{P}(x \in B) = \mathbf{P}(x \in A') = b) \\
&\geq \frac{b}{p}\Big[\inf_{x \in B} \eta(x) - \sup_{x \in A} \eta(x)\Big] \quad \text{(using } A' \subseteq A) \\
&\geq 0,
\end{aligned}
$$

where the last step follows from the definition of sets $A$ and $B$; in a similar manner, one can show that $\mathrm{TNR}_D[h'] - \mathrm{TNR}_D[h] \geq 0$.

For the case when $a < b$, we choose a set $B' \subset B$ with $\mathbf{P}(x \in B') = a$, and define $h'$ as

$$h'(x) = \begin{cases} 1 & \text{if } x \in B' \\ -1 & \text{if } x \in A \ . \\ h(x) & \text{o/w} \end{cases}$$

Similar to the previous case, one can show that $\mathrm{TPR}_D[h'] \geq \mathrm{TPR}_D[h]$ and $\mathrm{TNR}_D[h'] \geq \mathrm{TNR}_D[h]$.

In both these cases, we have by monotonicity of $\Psi$ that $\mathcal{P}_D^\Psi[h'] \geq \mathcal{P}_D^\Psi[h]$. Note that unless $a = b$, $h' \notin \mathcal{T}_\eta$. Hence, when $a \neq b$, we can view $h'$ as the new $h$, and apply the above exchange argument repeatedly to eventually arrive at $\widetilde{h} \in \mathcal{T}_\eta$ with $\mathcal{P}_D^\Psi[\widetilde{h}] \geq \mathcal{P}_D^\Psi[h]$, as desired. $\qquad\square$

## C   Example Distribution Where the Optimal Classifier for G-mean, H-mean and Q-mean is Not Threshold-based

We now present an example of a distribution under which the optimal classifier for the G-Mean, H-Mean and Q-Mean performance measures (see Table 1) is not of the requisite thresholded form, i.e. not of the form $\text{sign} \circ (\eta(x) - c)$ for any $c \in (0, 1)$.

Let $\mathcal{X} = \{x_1, x_2, x_3\}$. For some a constant $\epsilon \in (0, 1/2)$, consider the following distribution $D$ over $\mathcal{X} \times \{\pm 1\}$:

|       | $\mathbf{P}(x)$ | $\eta(x) = \mathbf{P}(y=1|x)$ |
|-------|------|----------------|
| $x_1$ | 0.25 | $1/2 - \epsilon$ |
| $x_2$ | 0.5  | $1/2$ |
| $x_3$ | 0.25 | $1/2 + \epsilon$ |

Table 4: Example distribution $D$ over $\mathcal{X} \times \{\pm 1\}$, where the optimal classifier for the G-mean, H-mean and Q-mean performance measures is not threshold-based.

Consider the following binary classifiers defined on $\mathcal{X}$:

$$\widetilde{h}_0(x) = \begin{cases} 1 & \text{if } x \in \{x_1, x_2, x_3\} \\ -1 & \text{o/w} \end{cases}$$

$$\widetilde{h}_1(x) = \begin{cases} 1 & \text{if } x \in \{x_1, x_2\} \\ -1 & \text{o/w} \end{cases}$$

$$\widetilde{h}_2(x) = \begin{cases} 1 & \text{if } x \in \{x_1\} \\ -1 & \text{o/w} \end{cases}$$

$$\widetilde{h}_3(x) = \begin{cases} -1 & \text{if } x \in \{x_1, x_2, x_3\} \\ 1 & \text{o/w} \end{cases}$$

$$h_4(x) = \begin{cases} 1 & \text{if } x \in \{x_2\} \\ -1 & \text{o/w} \end{cases},$$

where the first four classifiers constitute the set of all classifiers on $\mathcal{X}$ of the form $\text{sign} \circ (\eta - c)$ for $c \in (0, 1)$ (indicated by a '$\sim$'), while the last one is not of a thresholded form. We next list out in Table 5 the values of the G-mean, H-mean and Q-mean performance measures for these classifiers. It can be seen that for distributions defined using a small value of $\epsilon \in (0, 0.5)$, for each of G-Mean, H-Mean and Q-Mean, the classifier $h_4$ offers a higher performance measure value than any of the threshold-based classifiers. Clearly, threshold-based classifiers are not optimal under distributions of the above form with small values of $\epsilon$.

|  | TPR | TNR | G-Mean $\sqrt{\text{TPR} \cdot \text{TNR}}$ | H-Mean $2/\left(\frac{1}{\text{TPR}} + \frac{1}{\text{TNR}}\right)$ | Q-Mean $1 - \left((1 - \text{TPR})^2 + (1 - \text{TNR})^2\right)/2$ |
|---|-----|-----|-----------|---------|--------|
| $\widetilde{h}_0$ | 1 | 0 | 0 | 0 | 1/2 |
| $\widetilde{h}_1$ | $3/4 + \epsilon/2$ | $1/4 + \epsilon/2$ | $\sqrt{3}/4 + O(\sqrt{\epsilon})$ | $3/8 + O(\epsilon)$ | $11/16 + O(\epsilon^2)$ |
| $\widetilde{h}_2$ | $1/4 + \epsilon/2$ | $3/4 + \epsilon/2$ | $\sqrt{3}/4 + O(\sqrt{\epsilon})$ | $3/8 + O(\epsilon)$ | $11/16 + O(\epsilon^2)$ |
| $\widetilde{h}_3$ | 0 | 1 | 0 | 0 | 1/2 |
| $h_4$ | 1/2 | 1/2 | **1/2** | **1/2** | **3/4** |

Table 5: Performance measures G-mean, H-mean and Q-mean evaluated for classifiers $\widetilde{h}_0, \widetilde{h}_1, \widetilde{h}_2, \widetilde{h}_3$ and $h_4$ under the example distribution $D$ in Table 4. Here $\epsilon \in (0, 0.5)$. For small values of $\epsilon$, classifier $h_4$ offers the best value w.r.t. all measures (highlighted in bold).

## Footnotes

[5]We used the SVM$^{\text{perf}}$ routine provided in `http://www.cs.cornell.edu/people/tj/svm_light/svm_perf.html` for the $F_1$-measure; we made necessary modifications to this code (as prescribed in [12]) to optimize G-TP/PR and G-Mean.

[6]Here, we cannot set the regularization parameter to $1/\sqrt{|S|}$ since the theoretical prescriptions of [20] are not applicable to multivariate extension of hinge loss optimized by SVM$^{\text{perf}}$.

[7]Here, we make use of the fact that for any two sequences of non-negative random variables $X_n$ and $Y_n$ for which $X_n + Y_n \xrightarrow{P} 0$, we have $X_n \xrightarrow{P} 0$ and $Y_n \xrightarrow{P} 0$.