[Reviews · NeurIPS 2014]

Submitted by Assigned_Reviewer_4

The paper studies the statistical consistency of plug in classifiers under non decomposable loss functions such as the F statistic which is a popular performance measure in machine learning.

The problem studied in this paper is complex because non decomposable measures cannot, by definition, be expressed as an empirical expectation. Therefore, usual concentration inequalities are not applicable in this scenario.

The authors present a general analysis for measures that can be expressed as a continuous function of the true positive rate and the true negative rate as well as the class probability.

I have some qualms with the content of the paper. The main issue is that, in my opinion, this paper would be better fit for a statistics journal. Indeed, the main difference between machine learning and statistics is that we are interested in dealing with finite samples, therefore consistency results such as the ones presented in this paper do not necessarily belong in a conference such as NIPS. If the paper had presented a finite sample analysis a better case for acceptance could have been made.

Another issue appears in the proof of Lemma 5. In particular, the probability that the true positive rate (TPR) converges uniformly over all hypothesis \theta is not immediate, a more detailed explanation should be given to justify the convergence in probability for a union of uncountable events.

The rest of the results are simple applications of the main theorem to different non-decomposable measures. The experimental results are somewhat interesting, however there is no statistical significance analysis of the results, at the very least error bars representing the standard deviation of several replicas of the same experiment should be given.
Summary: The problem posed by the problem is interesting, however it seems more fit for a statistics conference. There are also some technical issues in the proof of a Lemma which is central for the rest of the theory given in the paper.

Submitted by Assigned_Reviewer_19

The paper studies statistical consistency properties of learning algorithms for non-decomposable performance measures. The authors analyze several measures of this type (balanced error rate, F-measure, G-TP/PR, G-mean, H-mean, Q-mean). The analysis concerns a plug-in classifier that tunes a threshold on a validation set. The authors prove the statistical consistency of this algorithm for two different assumptions. These results extend previous works on the statistical consistency for some of these measures, particularly the paper of Ye et al. (2012) which considers optimization of the F-measure. Moreover, the paper presents a general framework for proving the consistency properties. The paper contains also experimental results that confirm the theoretical findings and show that the analyzed approach is competitive in comparison to other state-of-the-art approaches (e.g., SVM-Perf).

This is a very well-written paper that extends our understanding of consistent learning algorithms. Although the paper is very theoretical, all the concepts are clearly introduced and the paper is easy to follow. Moreover, the obtained results are also significant from practical point of view as the performance measures studied by the authors are commonly used.

Minor comments:

- Assumption A and B: I would emphasize the difference between assumption A and B, as indeed for some of the performance measures the thresholding strategy is not always optimal.

- EUM vs. DTA: The authors could discuss in more detail the difference between the EUM (expected utility maximization) and DTA (decision-theoretic approach) frameworks. These two frameworks are two different formulations of the problem. The optimal solution of EUM does not necessarily correspond to the optimal solution for DTA.

After rebuttal:

I thank the authors for clarifying the main issues raised in the reviews.

Summary: This is a very well-written paper that contains very important results concerning consistency of thresholding algorithms for several non-decomposable performance measures. In my opinion, this paper should be accepted for NIPS.

Submitted by Assigned_Reviewer_38

Quality: Technical quality is very good, the theoretical results are sound and all the proofs are provided either in the main body of the paper, or in the appendix.

Clarity: Despite a strong emphasis on theoretical results (numerous lemmas and theorems), the paper was actually relatively easy to follow and the quality of writing is very good. Nevertheless, I think the paper is very heavy on notation, and this should be improved. I give some suggestions below, but I leave it to the authors what changes (if any) they prefer to include. For instance, I would suggest to drop subscript D in the whole paper, i.e. TPR_D, TNR_D, P_D, regret_D, etc., would be TPR, TNR, P, regret, etc. It obscures the notational, while the distribution is always the same (D) and clear from the context. It will also be easy to distinguish expected quantities from empirical quantities, as the latter are always indicated by the "hat" symbol. Moreover, it is probably not needed to explicitly denote the dependence on S_1 and S_2 all the time (e.g., by using \widehat{\eta}_{S_1} or \widehat{t}_{S_2,\widehat{\eta}_{S_1},\Phi}). In the same way, \Psi does not need to be used in almost all symbols, as it is always clear what \Psi do we mean (there is only one \Psi analyzed throughout the whole proof). I understand the authors wanted to be precise with notation, but I think it is obscure at the moment and has a negative effect on the readability of the paper.

Originality: As far as I am concerned, the results are novel and generalize some previous results shown for very specific cases (specific measure \Psi, stronger assumptions).

Significance: Judging by interesting theoretical results and good experimental verification, I think the paper should have an impact on the theory of learning with non-decomposable performance measures.

In my opinion, this is a good paper with interesting theoretical result. Although \Psi-consistency of L_1-consistent class probability estimator is not really that surprising, this is the first formal result of such generality for non-decomposable performance measures, and is worth publishing. In my opinion, the paper would benefit from answering to at least some of the questions below.
- Is assumption that the optimal threshold exists and is in (0,1) necessary? If yes, it would be nice to give an illustrative counterexample, which shows why consistency fails when the optimal threshold is in {0,1}, or when it does not exist. If not, it would be worth trying to prove the theorems without this assumption.
- Is Assumption A necessary (as compared to B) for non-monotonic performance measures? Again, I am missing an example of what can go wrong if \Psi is not monotonically increasing in TPR and TNR, while Assumption A is satisfied and Assumption B is not. If no such counterexample can be given, again, the authors should try to relax Assumption A.
- It would also be interesting to consider the case when probability estimates and threshold are obtained on the same data set S (without splitting into S_1 and S_2). This is a common learning scenario in the real life, and hence it is legitimate to ask under what conditions consistency holds in such a setting?

Some specific remarks:
- Line 185-186: what does it mean "eta(x) conditioned on y=1 and on y=-1 is mixed"?
- Line 192 (Theorem 1): "for some t^*_{D,\eta,\Psi} \in (0,1)" -- this symbol (t^*_{D,\eta,\Psi}) is already defined (what does the word "some" mean here), and it follows from Assumption A, that it is in (0,1), so this statement is unnecessary (I think).
Summary: Overall good paper, with interesting theoretical results. Well written and relatively easy to follow, but quite heavy on notation (should be improved).
Author Feedback
Author rebuttal: We thank the reviewers for the detailed comments. Below are responses to main comments.

Reviewer 1:

Difference between Assumptions A and B: Certainly, we can add a note emphasizing this.

EUM vs DTA: The plug-in algorithms in our work invariably fall under the EUM paradigm. Indeed the two paradigms are different; while in EUM, the goal is to maximize a non-decomposable function of the population TPR and population TNR (as is the case in our work), in DTA, one looks at the 'expected' performance measure computed on a set of 'n' examples, and is interested in the value of this expectation when n->infty. (EUM involves a non-decomposable function of expectations, while DTA involves an expectation of a non-decomposable function.) We will certainly make this clear in the main text.

Reviewer 2:

Notation: Thanks for helpful suggestions. We will certainly consider dropping subscripts whenever they are clear from context.

Optimal threshold in {0,1}: It is only for ease of exposition that we have focused on optimal thresholds in (0,1). Our consistency results easily extend to the general case of optimal thresholds in [0,1], where we will have to use a minor variant of Algo 1, which in addition to thresholded classifiers, also considers the 'all 1s' classifier. In particular, the current definition of the 'sign' function (where sign(u) = 1 if u > 0 and -1 o/w; see pg 3) used while defining a plug-in classifier does not allow us to construct the 'all 1s' classifier, and hence this classifier needs to be included separately in the algorithm. Thm 1 can now be suitably modified to work for the new algorithm by handling the 'all 1s' classifier as a separate case. We can certainly add a note on this in the final version.

Clarification on Assumptions: Assumption A requires the CDF of \eta conditioned on y=1 and on y=-1 to be continuous at the optimal threshold t*; under this, we show consistency of plug-in w.r.t. any perf measure that is a continuous function of TPR & TNR, and for which the optimal classifier is of a thresholded form. We apply this result to prove consistency for AM, F and G-TP/PR (Sec 4), where in each case, the specific functional form of these measures is used to show that the optimal classifier is of the requisite form (Lem 7-9). If we make a stronger assumption of continuity of the above CDFs over the entire range (0,1) (Assumption B), we can show consistency for any perf measure that is monotone in TPR & TNR *without having to appeal to the precise functional form of the measure* (Sec 5).

Assumption A is critical in proving convergence of TPR & TNR (and hence of any perf measure that is a continuous function of TPR & TNR) as it is required in Eq (8) in proof of Lem 2. Indeed one can construct discontinuous distributions where this convergence fails; e.g., this would be the case when the marginal distribution of D over the instances in X has a point mass at an instance 'x' for which \eta(x) = t*, resulting in the above CDFs having a 'jump' discontinuity at t*. We would like to stress however that Assumption A is quite mild as it requires continuity at only one point and is satisfied by a wide range of distributions.

Prob estimates and threshold from same data S: This would induce a dependence of the learnt prob estimation function on S, in which case one can no more appeal to convergence of probability estimates (via WLLN) in the proof of Lem 4 (which require iid draws of the given sample for a 'fixed' classifier h that is independent of the sample); a similar issue also arises with Lem 5. However, this is certainly an interesting question and a potential direction for future work.

Specific remarks:
(a) 'mixed' refers to a probability distribution that has both discrete and continuous parts
(b) thanks for pointing this

Reviewer 3:

Lem 5 (Uniform convergence for TPR): Sorry for missing out a few details in the proof of Lem 5 in Appendix. This result is indeed correct and follows from a standard VC-dimension based argument. In particular, noting that the VC-dim of the class of threshold classifiers is 1, a standard technique used to show a similar uniform convergence result for the 0-1 classification error (along with convergence of \hat{p} to p) can be used to derive this result for TPR/TNR. We will certainly include the complete proof in the final version.

Relevance to NIPS: We strongly believe that our paper is very relevant to NIPS. Statistical consistency is a fundamental property of any supervised learning algorithm, and has received enormous attention from the machine learning community in recent years (e.g. refs below), with papers on consistency receiving best paper/honorable mention awards in both NIPS and ICML (ref 1,2). While finite sample results are nice to have, asymptotic consistency results, such as the ones in our paper, are certainly interesting in their own right, and have been very well-received in the past in ML venues (e.g. refs below). Moreover, the performance measures studied in our paper are widely used in several ML applications, and hence consistency results for these measures would indeed be of great interest to the ML community.

1. Duchi et al. On the Consistency of Ranking Algorithms. ICML'10. (Best student paper)
2. Calauzenes et al. "On the (Non-)existence of Convex, Calibrated Surrogate Losses for Ranking". NIPS'12. (Honorable mention)
3. Ye et al. "Optimizing F-Measures: A Tale of Two Approaches". ICML'12.
4. Dembczynski et al. "Consistent Multilabel Ranking through Univariate Loss Minimization .." ICML'12.
5. Ramaswamy et al. "Convex Calibrated Surrogates for Low-Rank Loss Matrices .." NIPS'13.
6. Menon et al. "On the Statistical Consistency of Algorithms for Binary Classification under Class Imbalance". ICML'13.
7. Dembczynski et al. "Optimizing the F-Measure in Multi-Label Classification: Plug-in Rule .." ICML'13.

Error bars in expt: Thanks for the suggestion. These can certainly be included.